# Interventions to improve access to opioid agonist therapy in acute hospitals: A scoping review

Dan Lewer [1,2]*, Nerissa Tilouche [3], Molly Bradbury [4], Thomas D. Brothers [2,5,6],
Adam Holland [3,7], Vivian Hope [8], Rosalind Gittins [9], Jenny Scott [10], Kendall Searle [3],
Gareth Watson [11], Magdalena Harris [3]

1 Bradford Institute for Health Research, Bradford Teaching Hospitals NHS Foundation Trust, Bradford,
United Kingdom, 2 Department of Epidemiology and Public Health, University College London, London,
United Kingdom, 3 Department of Public Health, Environments and Society, London School of Hygiene
and Tropical Medicine, London, United Kingdom, 4 Severn Foundation School, Bristol, United Kingdom,
5 Department of Medicine, Dalhousie University, Halifax, Nova Scotia, Canada, 6 Addiction Medicine
Consult Service, Nova Scotia Health, Halifax, Nova Scotia, Canada, 7 School of Psychological Science,
University of Bristol, Bristol, United Kingdom, 8 Public Health Institute, Liverpool John Moores University,
Liverpool, United Kingdom, 9 Aston Pharmacy School, Aston University, Birmingham, United Kingdom,
10 Centre for Academic Primary Care, Bristol Medical School, University of Bristol, Bristol, United
Kingdom, 11 Guy's and St Thomas' NHS Foundation Trust, London, United Kingdom

☺ These authors contributed equally to this work.
* d.lewer@ucl.ac.uk

School of Medicine: NOSM University, CANADA

**Peer Review History:** PLOS recognizes the
benefits of transparency in the peer review
process; therefore, we enable the publication
of all of the content of peer review and
author responses alongside final, published
articles. The editorial history of this article is
available here: https://doi.org/10.1371/journal.
pmen.0000322

## Abstract

Many people who use illicit opioids have negative experiences when admitted to
hospital, which is partly due to poor availability of opioid agonist therapy (OAT). We
conducted a scoping review of interventions to increase access OAT to for hospital
patients, with searches of MEDLINE, EMBASE, PsychINFO, and CINAHL for evaluations published before 29 July 2024. We followed a registered protocol (identifier:
CRD42022313237). We included interventions in acute inpatient or emergency
department settings, and extracted intervention characteristics, location, evaluation
design and quality, and evidence for effectiveness. We included 57 studies; 50 from
the United States, six from Canada, and one from the UK. Fifty-one were published
in 2015 or later. We identified three intervention classes: (a) pathways to initiate OAT
in emergency departments, entailing screening patients or training staff to identify
withdrawal, initiating buprenorphine, and supported referrals (26 studies); (b) multidisciplinary 'addiction consult teams', which provide substance-related care across
hospital departments, advise primary medical teams on issues such as pain relief
and withdrawal management, and support patients with discharge and onward care
(18 studies); and (c) Interventions that build capacity of general clinical teams to
provide OAT to inpatients, including protocols to identify patients who need OAT, multidisciplinary patient review, and training/clinical education (13 studies). Most interventions included multiple components, and the most common were clinical education
and measures to improve continuity of OAT after discharge, such as bridge prescriptions and supported referrals to community prescribers. Almost all studies concluded

**Data availability statement:** All data are in the manuscript and/or supporting information files.

**Funding:** This work was supported by The National Institute for Health and Care Research (HSDR NIHR133022 to DL, NT, VH, RG, JS, and MH); the Dalhousie University Internal Medicine Research Foundation Fellowship (Canadian Institutes of Health Research Fellowship CIHR-FRN#171259 to TDB); the National Institutes of Health/National Institute on Drug Abuse (Research in Addiction Medicine Scholars Program R25DA033211 to TDB); the Canadian Institutes of Health (CIHR-PCS#185469 to TDB); Medical Research Council (GW4 BioMed2 Doctoral Training Partnership MR/W006308/1 to AH; Doctoral Training Fellowship MR/X018636/1 to AH). The funders had no role in study design, data collection and analysis, decision to publish, or preparation of the manuscript.

**Competing interests:** We have read the journal's policy and the authors of this manuscript have the following competing interests: AH is Co-Chair of the Faculty of Public Health Drugs Special Interest Group; he volunteers for the Loop (a drug checking organisation); and is a member of the Drug Science Enhanced Harm Reduction Working Group. JS is paid for clinical work undertaken with Turning Point, a drug and alcohol treatment service; and her academic post is part-funded by NOVA-21 (Novel Research to Advance HIV Prevention) RFP Program, Gilead. RG is a Scientific Committee member at Drug Science and is a member of the Medical Psychedelics and Enhanced Harm Reduction Working Groups; she is immediate past president of the College of Mental Health Pharmacy; she is Chief Pharmacy Officer of the General Pharmaceutical Council; she is senior trainer for SLD Training and she is lead for the RCGP accredited drug use course; and she is National Clinical Advisor for the Addiction Mission.

that interventions were effective, however evaluation methods were generally weak and most used before/after or case series designs. Efforts to improve OAT in acute hospitals emerged recently in North America and focus on addiction consult teams and initiation of buprenorphine in emergency departments. Although formal evaluation is weak, these models may represent starting points for national policy and larger research programmes.

## Introduction

Many people who use illicit opioids have negative experiences when admitted to hospital. Reasons include stigmatising attitudes among hospital staff, diagnostic overshadowing in which symptoms are attributed to drug use and not fully investigated, and poor pain relief [1–6]. Some hospital patients may have pre-existing opioid agonist therapy (OAT), such as methadone or buprenorphine [7]. These medicines are typically managed by community-based prescribers. For hospital patients who have OAT prescriptions, a key determinant of the quality of care is continuity of these medicines during a hospital admission [8]. Other patients who use opioids and do not have pre-existing OAT may benefit from initiation of OAT while they are in hospital. However, in many settings few eligible hospital patients receive OAT [9] and qualitative research suggests that OAT is often delayed, provided at a low dose, or not provided at all [3,9–11]. Patients may be aware of these issues and delay presentation at hospital, or leave hospital before treatment is complete (known as 'discharge before medically advised', 'patient-directed discharge' or 'discharge against medical advice') [11–13].

There have been many calls for hospitals to provide better care for patients with opioid dependence [14]. The issue of poor care and outcomes in this patient group has been recognised since the 1970s, including the complexities of pain relief and reconciliation of doses with community-based providers [15]. Some hospitals have established projects to improve the availability of OAT. It is difficult to estimate how many hospitals have addressed this issue because the results of quality improvement projects are often not published [16]. It appears that many hospitals do not have specific opioid withdrawal protocols, or these protocols do not follow the best evidence [17]. Improvement projects are usually established by clinicians with a special interest rather than as part of a wider policy [18]. In North America, an increase in opioid-related deaths [19] has led to calls for emergency departments to support patients to initiate OAT [20,21]. A body of research has evaluated initiation of buprenorphine in emergency departments, primarily as a method of increasing OAT coverage in the population, with outcomes focused on linkage to community-based OAT [22–25]. This research includes randomised trials of interventions such as screening, brief intervention and referral; and modified workflow in electronic health records to assist with screening for drug use and buprenorphine initiation [26–28]. A smaller body of research focuses on improving inpatient care for people who use drugs, with models including multidisciplinary addiction consult teams, clinicians from community-based

substance use services visiting hospitals to support patients, and training about substance dependence for general hospital clinicians [18,29,30].

Research in the UK has shown that hospital OAT policies are often unclear or include unnecessary procedural barriers such as the requirement for a positive urine opioid test before OAT is offered [17], and an international review of published hospital guidelines found they were often not evidence-based [31]. We did a scoping review of research into interventions in acute hospital settings that aimed to increase the proportion of eligible patients that receive OAT. Our review aimed to: (1) examine the extent of published evaluations of relevant interventions, (2) describe and classify interventions, (3) evaluate research quality, and (4) make recommendations for future research.

## Materials and methods

### Protocol

We conducted a scoping review and this report follows PRISMA-ScR (Preferred Reporting Items for Systematic reviews and Meta-Analyses extension for Scoping Reviews) [32]. Scoping reviews map the extent, methods, and key concepts in a research field [32,33]. They typically include studies with a broad range of objectives and methodologies, in contrast with traditional systematic reviews that aim to synthesize comparable results. We registered a protocol for this review with PROSPERO (identifier: CRD42022313237) [34].

### Search strategy

We searched MEDLINE, EMBASE, PsycINFO and CINAHL from inception until 29 July 2024 using keywords and MeSH terms related to hospital care and opioid agonist therapy (search terms for Medline are shown in the Box, with terms for other databases included in "Search Terms" in S1 Text). Citations were de-duplicated and uploaded to Covidence software. Among studies included, we conducted both manual 'backwards searches' of each article's reference list and 'forwards searches' of studies referencing the included study. We used Google Scholar for forwards reference searching. Titles and abstracts for potentially relevant studies identified from reference searching were screened in the same way as those from database searches. Where we identified a related systematic review (we identified four [29,30,35,36]), we also searched references of these articles.

---

**Box: Medline search terms**

1. opioid agonist therapy.ti,ab.

2. opioid agonist treatment.ti,ab.

3. opiate substitution therapy.ti,ab.

4. methadone.ti,ab.

5. buprenorphine.ti,ab.

6. medication assisted treatment.ti,ab.

7. medications for addiction treatment.ti,ab.

8. medication for opioid use disorder.ti,ab.

9. or/1–8

10. hospital.ti,ab.

---

11. inpatient.ti,ab.

12. emergency department.ti,ab.

13. accident and emergency.ti,ab.

14. admission.ti,ab.

15. surg$.ti,ab.

16. or/10–15

17. 9 and 16

18. 9 and (or/10–14)

### Inclusion and exclusion criteria

We included evaluations of interventions in acute hospital settings that: (1) aimed to increase the proportion of eligible patients that receive OAT; (2) used process-level outcomes such as the likelihood an eligible patient is prescribed OAT or staff knowledge and skills, or patient-level outcomes such as leaving hospital before medically advised or pain control; and (3) were available in English language. We had no restriction on the study design (we included both quantitative and qualitative studies), date of intervention or publication, or country of intervention.

We excluded studies that: (1) estimated the efficacy of OAT, such as those comparing outcomes for hospital patients who were given OAT against eligible controls who were not given OAT, or comparisons between different OAT medicines or modalities; (2) evaluated linkage to community-based OAT after hospital discharge and did not address OAT within hospitals; (3) were based on samples in which the majority of participants were not patients or staff at acute hospitals; (4) did not describe the key features of the intervention; or (5) included fewer than ten patients.

### Quality assessment

We used the mixed methods appraisal tool (MMAT) because of its utility for both qualitative and quantitative designs [37,38]. We excluded studies that did not meet both MMAT screening questions: 'Are there clear research questions?' and 'Do the collected data allow the research questions to be answered?'

### Screening and data charting

Two authors screened each title and abstract (DL, MB, TB, MH, AH, VH, RG, KS, GW contributed to this stage and two of these authors screened each article), with conflicts resolved in team meetings. We retrieved full texts for articles that passed screening and emailed corresponding authors where we could not access full texts. One author read each full text and assessed whether it met the inclusion criteria, then used a piloted data charting tool (see "Data charting tool" in S1 Text) to record details including the evaluation design, intervention location, OAT medication type, start date and duration of intervention, intervention description, intervention components, sample size, and outcomes. A second author then checked the data. The full dataset is provided in S1 Data.

### Synthesis and classification of interventions

Before collecting data, we developed an a-priori list of intervention components that we expected to find, based on a group discussion and review of literature known to the authors. Components included: (1) introduction of clinical guidance and protocols for OAT; (2) medicines reconciliation (confirming community-based OAT doses); (3) interventions to improve assessment of a patient's need for OAT, such as approaches to withdrawal measurement (e.g., the Clinical Opiate

Withdrawal Scale - COWS [39]) and biological testing to confirm recent drug use; and (4) non-pharmacological interventions to improve access to OAT, such as peer support and specialist liaison/consult teams.

We labelled interventions according to these components, with interventions allowed to include more than one label. Where relevant interventions included components that did not appear on the list, we updated the list. After data collection was complete, we created a mutually exclusive classification of interventions to describe general approaches to improving access to OAT in acute hospitals and described interventions in each class. Finally, we considered the relevance of the review findings in terms of research, policy, and practice.

## Results

### Search results

Database searches identified 7,702 unique articles and reference searching identified an additional 115. After screening titles and abstracts, we selected 217 articles for full text review (186 through the database search and 31 from reference searches). After full-text review, 58 studies met inclusion criteria, and we excluded one because it did not meet the MMAT screening criteria (Fig 1). The 57 included studies captured 48 unique interventions.

### Study characteristics

Study characteristics are summarised in Table 1. Most (50/57; 87.7%) evaluated interventions in the United States, of which the greatest number (20/50) were in northeastern states; 6/57 (10.5%) were in Canada, and 1/57 (1.8%) was

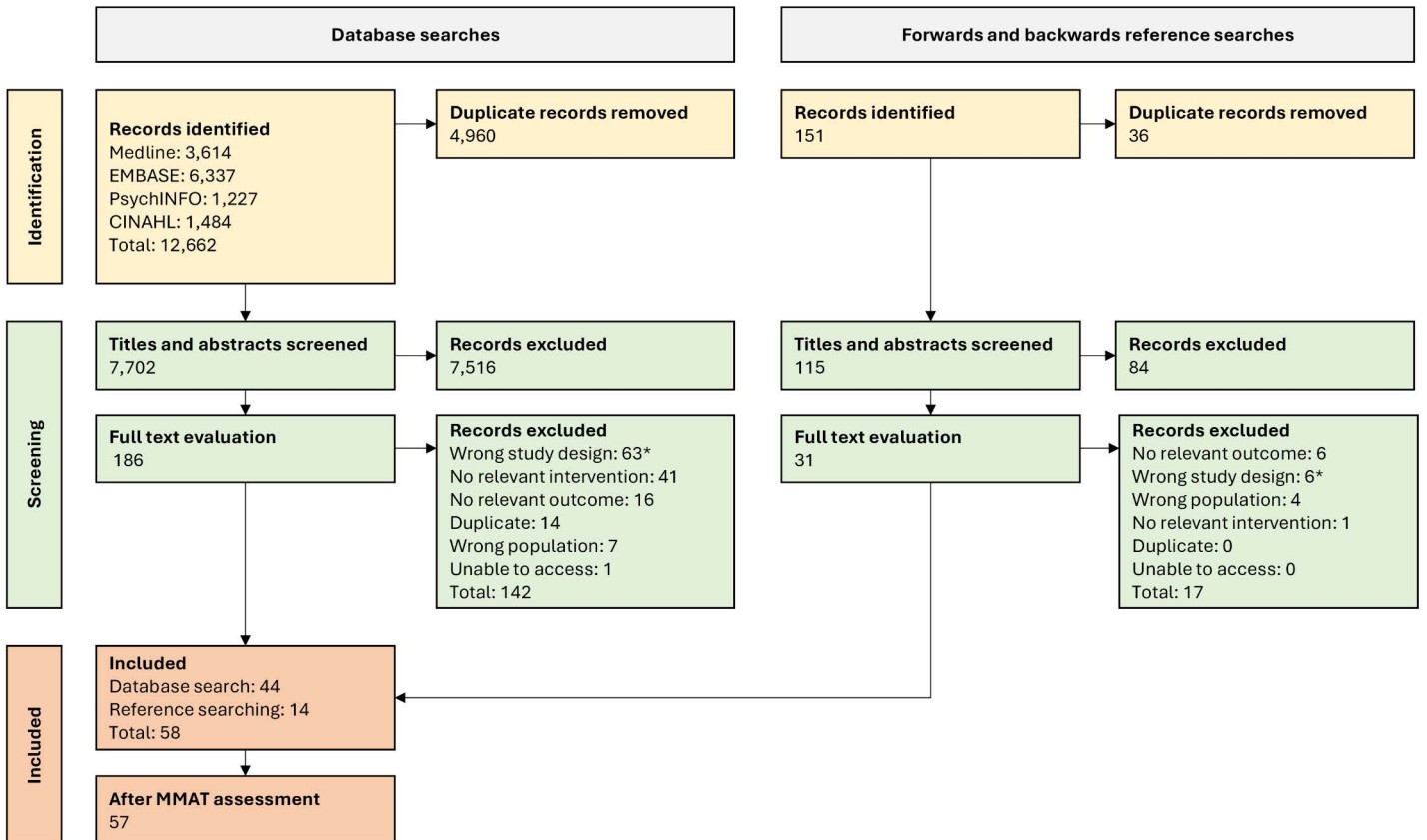

**Fig 1. Flow chart of study inclusion.** * Includes four systematic reviews – see "Changes from review protocol" in S1 Text.

**Table 1. Characteristics of studies.**

| Variable | Level | | Number | % |
|---|---|---|---|---|
| Total | | | 57 | 100.0 |
| Country of intervention | United States | Total | 50 | 87.7 |
| | | Northeast states | 20 | 35.1 |
| | | Western states | 10 | 17.5 |
| | | Southern states | 11 | 19.3 |
| | | Midwest states | 5 | 8.8 |
| | | Other/ nationwide | 4 | 7.0 |
| | Canada | Total | 6 | 10.5 |
| | | Ontario | 1 | 1.8 |
| | | Alberta | 2 | 3.5 |
| | | Nova Scotia | 1 | 1.8 |
| | | British Columbia | 1 | 1.8 |
| | | Other/ nationwide | 1 | 1.8 |
| | United Kingdom | | 1 | 1.8 |
| Study design | Before vs. after | | 26 | 45.6 |
| | Case series | | 13 | 22.8 |
| | Cohort study (with control group) | | 6 | 10.5 |
| | Qualitative | | 6 | 10.5 |
| | RCT | | 2 | 3.5 |
| | Time series | | 2 | 3.5 |
| | Cross-sectional (with control group) | | 1 | 1.8 |
| | Other | | 1 | 1.8 |
| Setting | Emergency department | | 26 | 45.6 |
| | Medical wards | | 16 | 28.1 |
| | Hospital-wide | | 15 | 26.3 |
| Primary medicine | Buprenorphine | | 36 | 63.2 |
| | Methadone and buprenorphine | | 18 | 31.6 |
| | Methadone | | 2 | 3.5 |
| | Unclear | | 1 | 1.8 |
| Intervention start date | 2000-2004 | | 1 | 1.8 |
| | 2005-2009 | | 1 | 1.8 |
| | 2010-2014 | | 3 | 5.3 |
| | 2015-2019 | | 44 | 77.2 |
| | 2020-2024 | | 7 | 12.3 |
| | No information | | 1 | 1.8 |

in the United Kingdom. Most studies were of recent interventions, with 51/57 (89.5%) interventions starting in 2015 or later.

## Study designs and quality assessment

The most common study design was a before/after comparison, used in 26/57 (45.6%) studies [25,40–64]. Examples included surveys of staff knowledge or attitudes before and after training [41,42,45,46,53,58,59,61] and changes in the proportion of patients with opioid dependence who were initiated on OAT before and after establishment of specialist substance use teams [45,48]. These studies did not account for background trends in the outcome or regression-to-the-mean

(a bias that occurs when an intervention is implemented when outcomes are unusual, and would have returned to 'normal levels' in the absence of the intervention). MMAT highlighted that these studies did not account for confounding, due to the absence of a control group.

The next most common design was a case series describing characteristics of patients receiving an intervention, used by 13/57 (22.8%) studies [65–77]. MMAT identified few problems with these studies, and they typically used appropriate methods for describing characteristics of a patient cohort, such as demographics or rates of OAT prescriptions. However, these studies provide limited evidence of effectiveness. For example, a case series of patients with drug-use associated endocarditis treated by a new multidisciplinary team found that 65% were discharged with OAT [66]. Authors concluded that the intervention was successful, though it is not possible to estimate an effect of the intervention using these data.

Six out of 57 studies (10.5%) used qualitative methods [78–83]. For two studies, MMAT identified discrepancies regarding the study description [80,82]: one study was described as 'focused ethnography' [80], but only used semi structured interviews and the other was labelled as a 'inductive' study but used a framework that appeared to be deductive [82].

Six out of 57 studies (10.5%) used cohort designs [84–89] to compare patients who received an intervention to a control group of patients that did not receive the intervention. These studies controlled some measured patient-level differences between exposure groups. The control groups were patients that were not selected to receive the intervention. This may mean the results are affected by confounding-by-indication, which occurs when certain types of patients are selected (or self-select) for an intervention, making the intervention and control groups different in ways that are not controlled by measured covariates. MMAT identified that these studies were susceptible to confounding. An exception was an evaluation of an intervention that involved screening patients in emergency departments for opioid withdrawal and initiating buprenorphine [86], which used a difference-in-differences design to control this type of confounding.

Two out of 57 studies (3.5%) were randomised trials [26,28]; one focusing on initiation and continuity of OAT from hospital to community [28] and the other on electronic workflow to support buprenorphine initiation in ED [26]. MMAT identified lack of blinding in both because placebo interventions were unfeasible.

Two out of 57 studies (3.5%) used time series methods to test whether there was a change in the rate of OAT initiations associated with an intervention [90,91]. One of these studies, evaluating multidisciplinary teams that initiate OAT in an Emergency Department [90], also included a time series from control hospitals that did not implement the teams, though did not formally estimate the effect of the intervention.

In 52/57 studies (91.2%), authors concluded that the intervention was effective. In 3/57 (5.3%) there were differing results across outcomes or purely descriptive aims. In the remaining 2/57 studies (3.5%), including the larger of the two randomised trials [26], authors concluded that the intervention was not effective.

## Intervention components

Our final list included nine intervention components, which were not mutually exclusive, and interventions could include multiple components. The components were (in order of frequency): (a) Measures to improve continuity after initiation of OAT in hospital (e.g., bridge prescriptions, partnerships with community OAT providers); (b) Training and education in relation to OAT, including 'X-waiver' training in the United States (prior to 2022, prescribers in the United States had to complete this training to gain a permit to prescribe buprenorphine); (c) Multidisciplinary patient review, typically entailing conferences in which professionals from different backgrounds review patients who use opioids and consider how access to OAT could be improved; (d) OAT guidance or protocols; (e) Providing advice about OAT to primary medical teams; (f) Peer support for patients who need OAT; (g) Electronic workflow that prompts staff to screen patients or supports other aspects of OAT; (h) Improving specialty-specific care for patients who are dependent on opioid (all evaluations in this group were focused on infections, with 4/5 studies focused on patients with infective endocarditis); and (i) Measures to improve continuity of pre-existing OAT (e.g., medicine reconciliation). Ten studies included one component only; nine

included two components; and 38 included three or more components, highlighting the multicomponent nature of most interventions in this field. Fig 2 summarises the frequency of these components within each class of intervention.

**Intervention classes**

We identified three discrete (mutually exclusive) classes of intervention: (a) pathways to initiate OAT in emergency departments; (b) addiction consult teams providing specialist support to other hospital teams; and (c) interventions that build capacity of general clinical teams to provide OAT to inpatients.

a.  Pathways to initiate opioid agonist therapy in emergency departments

We included 26 studies of interventions that aimed to increase initiation of OAT in emergency departments [25,26,40,41,43,44,47,58–63,67–69,71,72,74,81,82,86,90–93]. The most common model was screening patients for illicit opioid use and initiating buprenorphine while the patient was in the Emergency Department. Methods of identifying eligible patients included universal screening using questionnaires such as TAPS-1 [69] or DAST-10 [71], self-referral by patients, referral by peer navigators, and referral by clinicians. All interventions provided buprenorphine. The most common rationale for interventions in this class was that many people who use illicit opioids do not receive OAT in the community, and emergency departments are important contact points that could increase population coverage. Linkage to community-based OAT was typically an important feature of interventions and a primary outcome measure. A review of five case studies across the United States suggested that these interventions are often started by a 'champion' who is knowledgeable about opioid dependence and has existing relationships with local treatment services [81].

The most common components in these interventions were: (i) training about opioid dependence generally [41,43,47,58–61,68,74,91,93] and X-waiver training [44,47,93]; (ii) measures to improve continuity of OAT after initiation in hospital, especially 'bridge prescriptions' so that patients have take-home OAT to use between leaving hospital and accessing community-based OAT [74,81,92]; (iii) development of new guidelines or protocols [25,40,58,60,62,63,68,86,90,92,93]; and (iv) employment of peer navigators to provide non-clinical support including motivational interviewing and help with accessing community OAT and other health and social services [40,44,63,67,69,71,74,81,82,92,93]. Of the three classes, emergency department-focused interventions were most likely to use peer navigators. Some evidence suggests there can be bureaucratic barriers to paying peer navigators [81]. Five studies in this class evaluated training for emergency physicians and residents [41,43,47,59,61] and did not include other components; this was the largest group of single-component interventions.

b.  Addiction consult teams providing specialist support to patients and primary medical teams

We included 18 studies of addiction consult teams [42,52–54,64,66,70,73,75–80,83,84,88,89]. These interventions involved the establishment of a specialist in-house team that takes referrals from ward staff. Staffing of addiction consult teams varied, with the most common model being a multidisciplinary team including medical staff, nursing staff, and addictions-focused social workers [42,52,73,77–80,84]. Other models included medical residents (trainees) supervised by community-based addictions physicians [70] and trained doctors with mentorship from addictions specialists [50]. A key shared feature was that addiction consult teams responded to referrals from hospital staff, who identify patients that may benefit from OAT or other aspects of substance use care. The addiction consult team then assesses the patient with a focus on substance use. Provision of OAT is a core function, which may be prescribed directly by the addiction consult team, or by the primary medical team with advice. The addiction consult team may also provide harm reduction services (e.g., one study describes an addiction consult team providing sterile syringes and needles, naloxone kits, and sexual health screening [80]), assistance with access to addiction and social services in the community, and advice for ward staff on aspects of medical care that may be challenging for this patient group, such as pain control. No addiction consult team among those evaluated was available 24hrs; with some available in daytime hours only [52,78]. In one intervention, a new

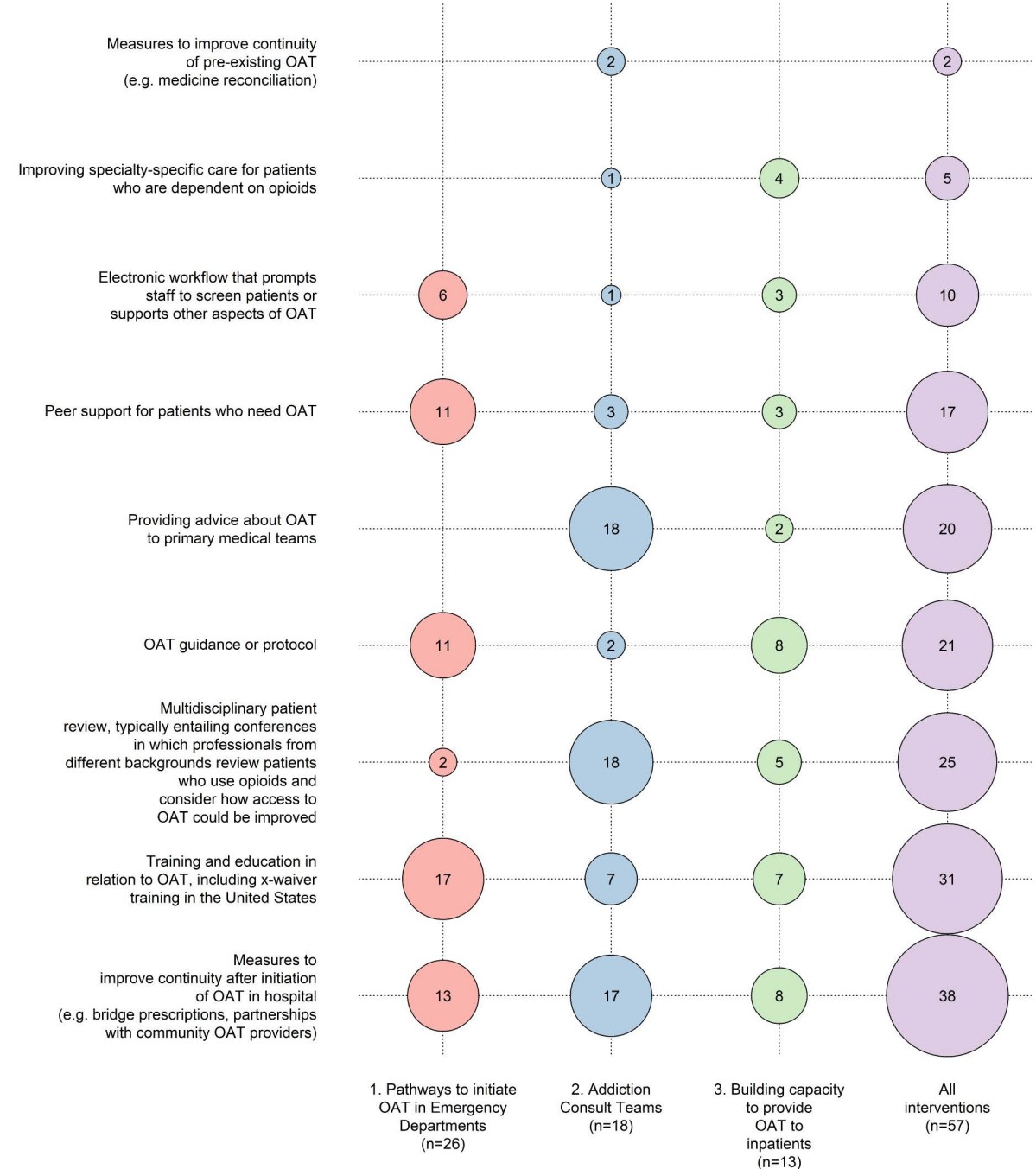

**Fig 2. Classification of studies that aim to improve access to opioid agonist therapy in acute hospital settings, and intervention components included in each study.**

team was established as a partnership between an existing addiction consult team and a local outpatient opioid treatment service [75]. The new team provided additional clinical support for OAT while patients were in hospital and organised linkage to outpatient treatment after discharge.

Addiction consult teams were multicomponent interventions. All included multidisciplinary patient review and advice to primary medical teams. Almost all included measures to improve continuity of OAT after hospital discharge. The interventions included different combinations of additional components, with some offering clinical education across the hospital [42,50,54,65,76,77,83] and others organising peer support for patients [42,76,89].

c. Interventions that build capacity of general clinical teams to provide OAT to inpatients

We identified 13 studies of interventions that build the capacity of teams not specialising in the management of opioid dependence to provide OAT for inpatients [28,45,46,48,49,51,53,55–57,66,85,87]. The common features of these interventions were a focus on inpatients and the development of systems, processes, and skills, rather than the establishment of a dedicated team. This class is more diverse than the first two and includes three main subclasses: (a) four interventions entailed a new hospital-wide protocol for screening all newly admitted patients for opioid dependence, screening for withdrawal, prescribing OAT where indicated, and referring to community services on discharge [45,49,56,57]. These interventions also included training for hospital staff, and one included modification of the electronic patient record to support clinical workflows [49]; (b) four interventions focused on improving care for patients with infections, of which three focused specifically on infective endocarditis [48,55,66,87]. These interventions are included in the review as they aimed to ensure OAT is provided for patients experiencing opioid dependence. The core feature of this subclass was multidisciplinary care conferences in which addiction specialists, infectious disease specialists, and other relevant specialties plan individual patient care. The interventions also typically included development of protocols addressing OAT provision alongside other issues including pain relief, and antibiotic management; and (c) five interventions focused on clinical education to improve awareness and access to OAT across the hospital [45,46,51,53,85]. Some interventions in this subclass were a discrete training module for hospital staff, focusing on OAT and related issues such as pain management for people who use opioids [45,46,53].

In the only study outside of North America, a hospital in the UK implemented a new protocol for managing opioid dependence, including clinical addictions assessments, methadone prescription, bridge prescriptions, and referral to community services [51]. This UK-based intervention was also the earliest to be implemented across the review, starting in 2004 (compared to a median start date of 2018).

## Discussion

We reviewed published evaluations of interventions aiming to increase access to OAT in acute hospitals. Most evidence is from North America and has been published since 2015, with most research focusing initiation of buprenorphine in emergency departments and addiction consult teams.

The dominance of North American research may reflect the recent opioid overdose crisis in this region, and the widespread availability of synthetic opioids such as fentanyl [94,95]. This crisis has led to a community of researchers seeking solutions, particularly in the United States. The recency of the evidence was also observed in a review of hospital-based harm reduction approaches such as needle and syringe programmes, which found that 90% of studies were from North America and 66% were published in 2019 or later [96]. Buprenorphine was the only medication option in emergency department interventions in the present review, which may in part be due to characteristics of the medication (such as safety in rapid titration) and in part due to federal legislation in the United States that limits methadone to specialised clinics [97,98] and prevents other options such as slow-release oral morphine.

Hospitals in North America likely face different problems to those in other countries. Relatively few eligible people in North America have OAT. One estimate suggests that 24% of people who inject drugs in Canada receive OAT and 19% in

the US [99,100]. This compares to 60% of people who use opioids in England [101] and similarly high rates in other Western European countries and Australia (low- and middle-income countries typically have much lower coverage) [99,100]. The relatively low coverage in North America may be explained by (a) the personal cost of OAT in the United States, (b) stigmatisation of OAT, (c) legal barriers to prescription, (d) the relative recency of initiation for many opioid users, and (d) potentially reduced efficacy of OAT in the fentanyl era [97,98,102,103]. The focus on OAT initiation in studies in the present review (see Fig 2) reflects the need to increase treatment coverage. Some evaluations of hospital-based interventions in North America exclude patients with pre-existing OAT due to their focus on initiation and expanding population access [89]. Hospitals in Western Europe and Australia would likely have more patients with pre-existing OAT and would therefore need a greater focus on continuity of doses rather than initiation, especially as medicine reconciliation is a common barrier to OAT provision in many hospitals [17].

Furthermore, the most prevalent illicit drugs vary substantially between countries and over time. For example, people in North America increasingly use synthetic opioids as opposed to heroin, which may be mixed with non-opioid drugs such as xylazine [104]. This means that higher doses of OAT may be required, and there could be greater benefits from short-acting opioids alongside non-opioid medications to relieve symptoms of withdrawal [105]. In the UK and Australia, most people who use illicit opioids still use heroin and most OAT prescriptions are methadone. While hospitals in these countries may be most concerned with continuity of methadone for patients who primarily use heroin, this situation may change with the proliferation of other drugs including benzodiazepine analogues, nitazene opioids, and xylazine [106–108].

The studies included in the present review covered diverse models, even within intervention classes. Addiction consult services had different staffing models and working patterns. Interventions to increase initiation of OAT in emergency departments varied more widely, with some focused on electronic workflow [25,26], some on peer support [69], some on training and education [41,43,47,59,61], and most including multiple components. The fact many interventions had multiple components may reflect a recognition of multiple barriers to good quality care for patients who use illicit opioids. Many included mechanisms to promote multidisciplinary working, training, and efforts to reduce stigma.

Although these models are emergent and formal evaluation is weak, these models may represent a starting point for national policy and larger research programmes. Implementation-focused research has highlighted barriers including stigma, fragmented policies, and funding [109,110]. Many interventions in this field depend on motivated 'champions' [81], and more sustainable interventions with robust evaluation may depend on national leadership and funding.

We are aware of four existing systematic reviews that addressed similar questions: (a) a review of emergency department-based interventions for people with opioid dependence, which included 12 studies, some of which estimated OAT efficacy [35]. This review found that most evidence was recent and from North America and concluded that stronger study designs are needed; (b) A review of interventions aiming to improve care for inpatients with opioid dependence, which identified 17 studies limiting to studies published from 2015-2020 [30]. This review found that most were delivered in a single healthcare system and focused on initiating OAT and linkage to post-discharge care; (c) A review of evidence relating to acute medical treatment of inpatients with opioid dependence, which identified 46 studies limiting to studies published from 2014-2019 [29]. This review included studies of OAT efficacy and non-OAT interventions such as management of infections among people who inject drugs, pain management, and overdose prevention. Authors concluded that acute hospitals need comprehensive addiction care, but the optimal components are still unclear; and (d) A review of evaluations of addiction consult teams, which identified 26 studies, all from the United States or Canada [36]. This review found that services included a variety of add-ons such as psychotherapy and discharge planning, and the evidence of effectiveness was weak due to a lack of control group in most studies. To our knowledge the present review is the first to focus on interventions that aim to improve access to OAT in acute hospital settings. Previous related reviews validate our findings that most evidence in this field is recent and from North America.

## Limitations of the evidence

First, we found that evaluation methods are weak. Few studies had well-defined control groups, and most use before/after or case series methods. This means it is difficult to draw general conclusions about the effectiveness of different approaches. Almost all evaluations concluded that interventions were effective at increasing access to OAT. Given methodological problems, this is likely to reflect multiple biases including regression to the mean, confounding by indication, multiple testing, and publication bias. Overall, this body of research provides insight into intervention forms but does not provide good evidence of effectiveness. Second, a scoping review maps existing research rather than the extent of activity. Few quality improvement projects are evaluated in peer-review articles [16], and there are likely many relevant interventions that are not captured in this review. This may also partly explain the lack of evidence outside of North America. In some countries OAT may be more 'mainstreamed' in hospitals and therefore subject to less research. Furthermore, the intervention features we describe are limited by the detail included in research reports. For example, we found that few studies described efforts to improve continuity of pre-existing OAT. This may reflect the focus of evaluations on initiation of OAT rather than a lack of this activity in hospitals in North America.

## Limitations of the review

First, we only included articles published in English language, which may partially explain the dominance of North American research. 698/12662 (5.5%) studies in our search were published in a language other than English and could not be included. Assuming these studies had the same probability of inclusion as the English-language studies, we would have included an additional 4 studies. In many cases articles in other languages had English titles and abstracts, though additional articles may not have appeared in the search due to the title and abstract not being in English. Second, our review only included acute hospital settings, and there may be useful evidence from other inpatient settings such as mental health hospitals. Third, we only included interventions that specifically focused on OAT, and therefore excluded interventions with a broader focus on alcohol and other substances in acute hospital settings that may have transferable findings. An example is a telelearning intervention for hospital staff that included modules on OAT alongside more general modules [111]. Fifth, we may have disproportionately excluded evidence related to quality improvement projects for two reasons: (a) we focused on peer reviewed publications and evaluations of quality improvement projects may be published as grey literature; (b) evaluations of quality improvement projects may be less likely to be translated in English.

# Conclusion

Efforts to improve OAT in acute hospitals emerged recently in North America and focus on addiction consult teams and initiation of buprenorphine in emergency departments. Although formal evaluation is weak, these models may represent starting points for national policy and larger research programmes.

# Supporting information

**S1 Text. Supplementary information.**
(PDF)

**S1 Data. Dataset.**
(CSV)

# Author contributions

**Conceptualization:** Dan Lewer, Magdalena Harris.

**Data curation:** Dan Lewer, Nerissa Tilouche.

**Formal analysis:** Dan Lewer, Nerissa Tilouche.

**Funding acquisition:** Dan Lewer, Vivian Hope, Rosalind Gittins, Jenny Scott, Magdalena Harris.

**Investigation:** Dan Lewer, Nerissa Tilouche, Molly Bradbury, Thomas D. Brothers, Adam Holland, Vivian Hope, Rosalind Gittins, Jenny Scott, Kendall Searle, Gareth Watson, Magdalena Harris.

**Methodology:** Dan Lewer, Nerissa Tilouche.

**Project administration:** Nerissa Tilouche.

**Supervision:** Magdalena Harris.

**Visualization:** Dan Lewer.

**Writing – original draft:** Dan Lewer, Nerissa Tilouche, Molly Bradbury, Thomas D. Brothers, Adam Holland, Vivian Hope, Rosalind Gittins, Jenny Scott, Gareth Watson, Magdalena Harris.

**Writing – review & editing:** Dan Lewer, Nerissa Tilouche, Molly Bradbury, Thomas D. Brothers, Adam Holland, Vivian Hope, Rosalind Gittins, Jenny Scott, Kendall Searle, Gareth Watson, Magdalena Harris.

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
