## [Decision Letter · Decision Letter 0]

3 Mar 2025

PMEN-D-25-00050

Interventions to improve access to opioid agonist therapy in acute hospitals: a scoping review

PLOS Mental Health

Dear Dr. Lewer,

Thank you for submitting your manuscript to PLOS Mental Health. After careful consideration, we feel that it has merit but does not fully meet PLOS Mental Health’s publication criteria as it currently stands. Therefore, we invite you to submit a revised version of the manuscript that addresses the points raised during the review process.

We look forward to receiving your revised manuscript.

Kind regards,

Craig Nicholas Cumming

Academic Editor

PLOS Mental Health

Journal Requirements:

https://journals.plos.org/mentalhealth/s/figures

https://journals.plos.org/mentalhealth/s/figures#loc-file-requirements 

Additional Editor Comments (if provided):

The reviewers have raised some valid points with respect to the ethics and robustness (primarily relating to the low response rate) of the survey component of your review, as well as its novelty, given the similar literature already published. Please pay particular attention to these concerns in your response to the reviewers comments, as this will help us assess this manuscript for suitability for publication.

Reviewers' comments:

Reviewer's Responses to Questions

**Comments to the Author**

1. Does this manuscript meet PLOS Mental Health’s publication criteria ? Is the manuscript technically sound, and do the data support the conclusions? The manuscript must describe methodologically and ethically rigorous research with conclusions that are appropriately drawn based on the data presented.

Reviewer #1: Partly

Reviewer #2: Yes

2. Has the statistical analysis been performed appropriately and rigorously?

Reviewer #1: Yes

Reviewer #2: N/A

3. Have the authors made all data underlying the findings in their manuscript fully available (please refer to the Data Availability Statement at the start of the manuscript PDF file)?

Reviewer #1: Yes

Reviewer #2: Yes

4. Is the manuscript presented in an intelligible fashion and written in standard English?

Reviewer #1: Yes

Reviewer #2: Yes

5. Review Comments to the Author

Reviewer #1: Some editorial issues:

- Instead of a space between end of sentence and citation [#], there is a fullstop or comma. Might be just a file conversion issue.

- Emergency Departments = make it lower case throughout

- They say "Two authors" did title and abstract screening but then say "(two of DL, MB, TB, MH, AH, VH, RG, KS, GW)" i.e. unclear who did what

Substantive issues:

- Overall I found the scoping review to be well done and well reported.

- p. 5 “...known as ‘patient directed discharge” – also include the more common previously accepted term “against medical advice”

- Could the search terms be put into a table or textbox instead of supplementary?

MAIN ISSUE:

- I was taken aback by the inclusion of a questionnaire to corresponding authors, as I have not seen a scoping review have an add-on qualitative question component.

(1) I didn't see evidence of ethics approval for this qualitative component/study. This is a critical issue.

(2) While the authors say "All authors consented to publication of these responses" the process of consent is unclear.

(3) The authors say that they described themes in the responses to these questions. However, they did not describe how they attended to rigour nor did they describe an analytical approach. These are critical components.

(4) They indicate they received responses from 16/57 authors – which is a fairly poor response rate.

(5) Overall, the most significant issue (beside no ethics statement) is what does this qualitative component add?

I recommend submitting the qualitative component as a separate study.

Reviewer #2: Thanks for the opportunity to review this scoping review of intervention to improve OAT initiation in acute care settings. The topic is important, and the paper is well written. That said, as the authors state, there are already four systematic reviews on similar topics, which somehow diminishes the novelty of the review.

In addition, their inclusion of a survey to authors about the experience with implementation which could have added an interesting and different angle is limited by the fact that they got less than one third of responses.

Another limitation of the present review, as the authors acknowledge, is that they only included articles written in English. This may have significantly biased their results towards only finding studies from North America. This is particularly problematic since as the authors also acknowledge studies from other settings where OAT coverage is higher (I.e., Western Europe) can provide some helpful insights to improve OAT coverage in North America.

Another limitation of only including studies in English language is that probably most of QI projects wouldn’t be translated to English and likely missed. Likewise, not searching for grey literature could also have resulted in having missed some key QI projects.

6. PLOS authors have the option to publish the peer review history of their article (what does this mean? ). If published, this will include your full peer review and any attached files.

**Do you want your identity to be public for this peer review?** For information about this choice, including consent withdrawal, please see our Privacy Policy .

Reviewer #1: No

Reviewer #2: No

---

## [Decision Letter · Decision Letter 1]

9 Apr 2025

PMEN-D-25-00050R1

Interventions to improve access to opioid agonist therapy in acute hospitals: a scoping review

PLOS Mental Health

Dear Dr. Lewer,

Thank you for submitting your manuscript to PLOS Mental Health. After careful consideration, we feel that it has merit but does not fully meet PLOS Mental Health’s publication criteria as it currently stands. Therefore, we invite you to submit a revised version of the manuscript that addresses the points raised during the review process.

We look forward to receiving your revised manuscript.

Kind regards,

Craig Nicholas Cumming

Academic Editor

PLOS Mental Health

Additional Editor Comments (if provided):

I tend to agree with Reviewer 1 that in light of the limitations of the author responses to the questionnaire that it adds limited value to the rest of the review, so I suggest removing this from your submission.

Please also address the other comments from Reviewer 1 so that we can consider your responses when making our decision about potential publication of your manuscript.

Reviewers' comments:

Reviewer's Responses to Questions

**Comments to the Author**

1. If the authors have adequately addressed your comments raised in a previous round of review and you feel that this manuscript is now acceptable for publication, you may indicate that here to bypass the “Comments to the Author” section, enter your conflict of interest statement in the “Confidential to Editor” section, and submit your "Accept" recommendation.

Reviewer #1: (No Response)

Reviewer #2: All comments have been addressed

2. Does this manuscript meet PLOS Mental Health’s publication criteria ? Is the manuscript technically sound, and do the data support the conclusions? The manuscript must describe methodologically and ethically rigorous research with conclusions that are appropriately drawn based on the data presented.

Reviewer #1: Partly

Reviewer #2: Yes

3. Has the statistical analysis been performed appropriately and rigorously?

Reviewer #1: N/A

Reviewer #2: N/A

4. Have the authors made all data underlying the findings in their manuscript fully available (please refer to the Data Availability Statement at the start of the manuscript PDF file)?

Reviewer #1: Yes

Reviewer #2: Yes

5. Is the manuscript presented in an intelligible fashion and written in standard English?

Reviewer #1: Yes

Reviewer #2: Yes

6. Review Comments to the Author

Reviewer #1: Need for editing:

Line 79: OAT [9]. and

Line 135: pain control; AND (3) were

Line 136: "We had no restriction on the study design (we included both quantitative and qualitative studies) should read "We included both quantitative and qualitative designs"

Lind 145: "We used the mixed methods appraisal tool (MMAT), which is an appraisal tool for reviews that..." should read "We used the mixed methods appraisal tool (MMAT) because of its utility for both qualitative and quantitative designs"

Line 161: "this list was" should read "COmponents included:

Line 165: drug use' AND (4)

Line 168: Very unclear sentence" We did not limit to interventions including these components and updated the list of components as we collected data."

Method:

Line 122 "Citations were de-duplicated and uploaded to Covidence"; why were all articles not uploaded, then de-duplicated to demonstrate the management of duplicates?

I am unsure whether the MMAT is the best instrument to individually assess Qual and Quant designs, as MM focuses on integration

I remain opposed to the inclusion of author responses. The qualitative component is biased by self-selection and response bias. 

Reviewer #2: (No Response)

7. PLOS authors have the option to publish the peer review history of their article (what does this mean? ). If published, this will include your full peer review and any attached files.

**Do you want your identity to be public for this peer review?** For information about this choice, including consent withdrawal, please see our Privacy Policy .

Reviewer #1: No

Reviewer #2: No

---

## [Editor Report · Decision Letter 2]

6 May 2025

Interventions to improve access to opioid agonist therapy in acute hospitals: a scoping review

PMEN-D-25-00050R2

Dear Dr Lewer,

We are pleased to inform you that your manuscript 'Interventions to improve access to opioid agonist therapy in acute hospitals: a scoping review' has been provisionally accepted for publication in PLOS Mental Health.

Best regards,

Kristen A Morin

Academic Editor

PLOS Mental Health
